# Analysis and Comparison of Proteomics of Placental Proteins from Cows Using Different Proteases

**DOI:** 10.3390/ani13213395

**Published:** 2023-11-01

**Authors:** Liuhong Shen, Zeru Zhang, Yue Zhang, Yuquan Zhao, Lei Fan, Shumin Yu, Suizhong Cao, Yixin Huang

**Affiliations:** The Key Laboratory of Animal Disease and Human Health of Sichuan Province, The Medical Research Center for Cow Disease, College of Veterinary Medicine, Sichuan Agricultural University, Chengdu 611130, China; 19834287107@163.com (Z.Z.); zhangyuesicau@163.com (Y.Z.); zhaoyq2728@163.com (Y.Z.); fanl9908@163.com (L.F.); yayushumin@sicau.edu.cn (S.Y.); suizhongcao@126.com (S.C.)

**Keywords:** dairy cows, placenta, trypsin, papain, pepsin, label-free

## Abstract

**Simple Summary:**

As an organ of fetal development, the placenta has many biological functions such as anti-oxidation, anti-tumor, anti-apoptosis, immune regulation, and skin care. It was found that the protein content, trace elements, and amino acid types of bovine placenta were similar to those of human placenta—which has been widely studied, developed, and utilized—and at the same time, the waste of resources caused by the random disposal of placenta was avoided. In this study, we explored and compared the proteome of different proteases following their action on cow placenta based on marker-free mass spectrometry and performed a bioinformatics analysis, which provided valuable reference information for the subsequent proteomic analysis of pregnancy-related diseases in cows, the study of the physiological functions of cow placenta, and the development and utilization of its by-products.

**Abstract:**

Newly found biochemical characteristics of the placenta can provide new insights for further studies on the possible markers of physiological/pathological pregnancy or the function of the placenta. We compared the proteome of the dairy cow placenta after enzymatic hydrolysis by three different proteases using a label-free mass spectrometry approach. In total, 541, 136, and 86 proteins were identified in the trypsin group (TRY), pepsin group (PEP), and papain group (PAP). By comparing the proteome of the PAP and TRY, PEP and TRY, and PEP and PAP groups, 432, 421, and 136 differentially expressed proteins were identified, respectively. We compared the up-regulated DEPs and down-regulated DEPs of each comparison group. The results show that the proteins identified by papain were mostly derived from the extracellular matrix and collagen, and were enriched in the relaxin signaling pathway and AGE-RAGE signaling pathway in diabetic complications; pepsin digestion was able to identify more muscle-related proteins, which were enriched in the lysosome, platelet activation, cardiac muscle contraction, the bacterial invasion of epithelial cells, and small cell lung cancer; trypsin mainly enzymatically degraded the extracellular matrix, blood particles, and cell-surface proteins that were enriched in arginine and proline metabolism, olfactory transduction proteasome, protein processing in the endoplasmic reticulum, pyruvate metabolism, and arrhythmogenic right ventricular cardiomyopathy (ARVC). In summary, these results provide insights into the discovery of the physiological functions of dairy cow placenta and the selection of proteases in dairy cow placenta proteomics.

## 1. Introduction

The placenta is an organ that connects the mother and fetus to maintain a stable environment for the growth and development of the fetus; it regulates the growth and development of the fetus by regulating the supply of nutrients, gases, hormones, etc., and has substance exchange, hormone secretion and barrier effects [1]. The placenta changes the mother’s endocrine and immune systems, establishing a blood vessel link between the mother and the embryo, which is the key to maintaining the growth of the fetus [2]. Abnormal placental function is one of the principal causes of fetal death. Proteomic technology is a common tool for studying placental abnormalities; it can be used to study changes in placental proteins during the disease process, clarify its pathogenesis, and identify differentially expressed proteins (DEPs) related to its pathogenesis—thereby providing an effective means for the clinical diagnosis of the disease. Placental proteomics can be used to identify more peptides and related biomarkers [3]. Using placental proteomics, differentiated proteins in human pre-eclamptic placentae were found to be closely related to mitochondrial function, indicating that mitochondrial dysfunction is a precursor of the pathogenesis of epilepsy [4]. Using placental proteomics, the occurrence of recurrent miscarriages in humans was found to be closely related to the core factors of early embryonic development, such as angiotensinogen, mitogen-activated protein kinase 14 (MAPK14), and prothrombin (F2) [5]. The digestion and extraction of protein using proteases is an important factor affecting the results of proteomics analyses. Presently, trypsin is often used in placental proteomics for protein extraction. Trypsin has a high specificity and only cleaves peptides at arginine and lysine residues [6]. However, some proteins that lack R and K residues cannot be digested by trypsin. In addition, when the trypsin cleavage site is located after a glycosylated asparagine, the attached carbohydrates may sterically hinder trypsin cleavage [7]. To overcome the limitations of trypsin use, non-specific enzymes such as papain and pronase—which can completely digest the protein—have been used. Trypsin is most effective in a neutral or slightly alkaline environment, but some proteins have low solubility in this environment and cannot be digested. Pepsin has the strongest activity in an acidic environment and can enzymatically hydrolyze proteins with greater solubility in an acidic environment. In a previous study, based on the Box–Behenken central response method, we established the preparation conditions for the highest reducing activity and extraction rate of trypsin, pepsin, and papain-hydrolyzed dairy cow placental peptides [8]. In this experiment, the products of protease-hydrolyzed cow placenta were analyzed using label-free technology. Qualitative and quantitative analyses of the protein and peptides obtained from cow placenta were compared to the biological information of cow placenta hydrolyzed by different proteases, which provides theoretical support for the study of protease selection in cow placenta proteomics.

## 2. Materials and Methods

### 2.1. Placenta Collection and Preparation

Sample collection was performed in strict accordance with the guidelines on the Care and Use of Laboratory Animals in China. All procedures were approved by the Animal Care and Use Committee of Sichuan Agricultural University. Placental samples were collected from 9 healthy pregnant Chinese Holstein cows in a large-scale, semi-closed unified farm in Sichuan province. Cows were cared in compliance with the Care and Use of Laboratory Animals of China, and all procedures were approved by the Animal Care and Use Committee of Sichuan Agricultural University. The cows were selected according to the following criteria: aged between 3 and 5 years, body weight above 600 kg, 2–4 parity, and around 40 weeks of pregnancy. We collected placental samples immediately after placentas had fallen off naturally. Only similar parts of the placentome were used for further analysis. Placental samples were washed in cold saline, portioned, frozen, and stored at −20 °C.

Placenta extracts were prepared according to the method reported by Shen et al. [9]. The placenta was homogenized using a homogenizer (Yuexin, Changzhou, China) in ultrapure water, then divided into three groups containing three placental homogenizations from nine cows. The homogenates were enzymatically digested under the following optimal conditions [8]: trypsin (Solarbio, 920T041), pepsin (Solarbio, 810H021), and papain (Solarbio, 621G025); enzymatic dissociation at reaction time 5.80 h, 4.70 h, and 5.49 h; substrate concentration 34.96%, 34.03%, and 35.74%; and enzyme base ratio 3.33%, 3.66%, and 3.92%, respectively. Each enzymatic hydrolysis was repeated three times. The hydrolysis samples were respectively marked as the trypsin group (TRY, which included trypsin-1, trypsin-2, and trypsin-3), the papain group (PAP, which included papain-1, papain-2, and papain-3), and the pepsin group (PEP, which included pepsin-1, pepsin-2, and pepsin-3). After digestion, homogenates were boiled for 10 min. Subsequently, the solution was cooled to room temperature. Then, the solution was centrifuged using a 5427 R centrifuge (Eppendorf, Hamburg, Germany) at 6000 rpm/min for 5 min at 4 °C. The supernatant was collected and freeze-dried (Telstar, Barcelona, Germany) for 24 h. The BCA Protein Assay Kit (Solarbio, PC0020) was used to measure the protein concentrations in a microplate and readings were taken at 562 nm on a NanoDrop OneC Microvolume UV-Vis Spectrophotometer (Thermo Fisher Scientific, Waltham, MA, USA). Then, the samples were dissolved in 21 μL MilliQ water containing 0.1% (*v*/*v*) formic acid for liquid chromatography with tandem mass spectrometry (LC–MS/MS) analysis.

### 2.2. Liquid Chromatography (LC)-Electrospray Ionization (ESI) Tandem MS (MS/MS) Analysis

Each fraction was injected for nano LC–MS/MS analysis. The peptide mixture was loaded onto a reverse phase trap column (Thermo Fisher Scientific, Waltham, MA, USA) connected to a C18 reverse phase analytical column (Thermo Fisher Scientific, Waltham, MA, USA) in buffer A (0.1% formic acid) and separated using a linear gradient of buffer B (84% acetonitrile and 0.1% formic acid) at a flow rate of 300 nL/min, controlled using intelliFlow technology. A two-hour gradient procedure was used as follows: 0–55% buffer B for 110 min, 55%–100% buffer B for 5 min, and 100% buffer B for 5 min. Then, a Q Exactive mass spectrometer (Thermo Fisher Scientific, Waltham, MA, USA) was used for LC–MS/MS analysis. The mass spectrometer was operated in positive ion mode. The scanning range of the precursor ion was 300–1800 *m*/*z*. The automatic gain control target was set to 1 × 10^6^ and the maximum injection time to 50 ms. The dynamic exclusion duration was 60 s. Survey scans were acquired at a resolution of 70,000 at 200 *m*/*z* and the resolution for the higher-energy C-trap dissociation spectra was set to 17,500 at 200 *m*/*z*, and the isolation window was 2 m/z. Moreover, the normalized collision energy was 30 eV and the underfill was 0.1%.

### 2.3. Label Free Analysis

The MS data were analyzed using Max Quant (version1.5.3.17, Max Planck Institute of Biochemistry, Martinsried, Germany). The MS data were searched against the UniProt *Bos taurus* database. The following parameters were considered relevant: enzyme, trypsin/papain/pepsin; max missed cleavages, 2; fixed modification, carbamidomethyl; variable modification, oxidation; main search, 6 ppm; first search, 20 ppm; MS/MS tolerance, 20 ppm; database pattern, reverse; include contaminants, true; peptide false discovery rate (FDR) ≤ 0.01; protein FDR ≤ 0.01; peptides used for protein quantification, razor and unique peptides; time window, 2 min; protein quantification, LFQ/iBAQ; and min ratio count, 1.

### 2.4. Bioinformatics Analysis

Functional annotation and network analysis were performed using STRING (URL accessed on 20 August 2022 at: http://string-db.org/) and Cytoscape platform version 3.8.2 (URL accessed on 25 August 2022 at: https://cytoscape.org/) based on *Bos taurus* genes. In particular, the two plugins of Cytoscape, namely ClueGo (version 2.5.7) and CluPedia (version 1.5.7), were used to integrate the Gene Ontology (GO) categories (biological process (BP), molecular function (MF), and cellular component (CC)), Reactome Pathways, Kyoto Encyclopedia of Genes and Genomes (KEGG), and Wiki Pathways annotation [10]. The κ score level was set at ≥0.4 while minimum and maximum levels were set at 3 and 8, respectively.

## 3. Results

### 3.1. Comparative Evaluation of Protein Extraction Efficiency in Cow Placenta Hydrolyzed by Three Proteases

The number of proteins identified from cow placenta using different digestion protocols differed significantly (425, 114, and 56 in TRY, PEP, and PAP, respectively; Table 1). A comparison of the digestion efficiencies of the three proteases showed that TRY had a significantly higher number of protein identifications than PEP and PAP. TRY extracted 3.7-times (*t*-test, *p* = 0.00132) and 7.5-times (*t*-test, *p* = 0.00015) more proteins than PEP and PAP, respectively. Moreover, PEP extracted two-times more proteins than PAP (*t*-test, *p* = 0.00678). A comparison of common proteins among the biological replicates of TRY and PEP revealed an overlap of 72–82%, while that of PAP ranged from 53% to 77% (Figure 1A). TRY and PEP had a higher reproducibility of acquisitions than PAP, with coefficients of variation in the range of 2–4% for proteins and 8–10% for peptides, which were lower than those for PAP (15% for proteins and 20% for peptides; Table 1). TRY, PEP, and PAP extraction resulted in 541, 136, and 86 quantifiable proteins (with LFQ intensity >0) from cow placenta, respectively. An analysis of the quantifiable proteins of cow placenta among the three protease groups showed that the numbers of common proteins were 80 (12.3%), 22 (3.4%), and 27 (4.1%) between TRY and PEP, PEP and PAP, and PAP and TRY, respectively. There were 449 (69.2%), 49 (7.6%), and 52 (8%) unique proteins in TRY, PEP, and PAP, respectively (Figure 2A). The common proteins constituted only a minority of the total quantifiable proteins, while the unique proteins of TRY constituted the largest proportion.

### 3.2. Analysis of Cow Placenta Quantifiable Proteins with Three Proteases

The quantifiable proteins of cow placenta extracted using trypsin, pepsin, and papain were further analyzed in terms of the distribution of the proteins’ molecular weight (Mw), as well as the sequence coverage, peptide length, and unique peptides ratio. Mw distribution is important in evaluating protein size. In this study, the Mw distribution of quantifiable proteins extracted using the three proteases was relatively wide and showed almost no difference. Mw findings demonstrated that 71%, 76%, and 67% of proteins in TRY, PEP, and PAP, respectively, weighed < 70 kDa. Meanwhile, approximately 20% of the quantifiable proteins weighed > 100 kDa (Figure 2A). Sequence coverage determined the overall accuracy of protein detection, which was relatively high in the present study. Moreover, 46%, 63%, and 42% of proteins in TRY, PEP, and PAP, respectively, had >10% sequence coverage distributions of quantifiable proteins (Figure 2B). Peptide lengths revealed the characteristics of the proteases. In the present study, 92%, 76%, and 88% of the detected peptides had a peptide length < 20 in TRY, PEP, and PAP, respectively (Figure 2C). Furthermore, each group consisted of a unique set of peptides, which endowed it with specific properties. Protein detection reliability tends to improve with the number of unique peptides in a protein group [11]. In the present study, the distribution curve of the number of unique peptides gradually increased, which indicates that the number of both the unique peptides and the reliable proteins was relatively high (Figure 2D).

### 3.3. GO and KEGG Analysis of Cow Placenta Quantifiable Proteins with Three Proteases

The GO analysis of cow placenta quantifiable proteins extracted using the three proteases showed highly similar distributions in terms of biological progress, cellular components, and molecular function (Figure 3A). The highest percentages were in metabolic progress and biological regulation, followed by cellular component organization, response to stimulus, and developmental process in biological progress. The quantifiable proteins were mainly distributed in the membrane, nucleus, protein-containing complexes, cytosol, and cytoskeleton in the cellular component. The molecular function-based analysis showed that a majority of the proteins were involved in processes such as protein binding, ion binding, hydrolase activity, nucleotide binding, structural molecule activity, and nucleic acid binding.

A KEGG pathway analysis was performed to understand the biological functions and specific pathways related to cow placenta. The top 10 KEGG pathways in TRY, PEP, and PAP are presented in Figure 3B. Focal adhesion, PI3K-Akt signaling pathway, human papillomavirus infection, and extracellular matrix (ECM)–receptor interaction were the common pathways in TRY, PEP, and PAP.

### 3.4. Identification of Differentially Expressed Proteins (DEPs)

The DEPs were defined based on a 2.0-fold change threshold (with a fold change > 2.0 or <0.50, *p* < 0.05; Figure 4) or specific expression (Table 2) in comparisons between groups using the mass spectra data. A comparison between PAP and TRY identified 432 DEPs, including 34 up-regulated and 398 down-regulated proteins (Figure 4A and Table 2). A comparison between PEP and TRY identified 421 DEPs, including 56 up-regulated and 365 down-regulated proteins (Figure 4B and Table 2). A comparison between PEP and PAP detected 136 DEPs, including 100 up-regulated and 36 down-regulated proteins (Figure 4C and Table 2).

### 3.5. Cluster Analysis of DEPs

The hierarchical clustering algorithm (Hierarchical Cluster) was used to perform a cluster analysis on each group of DEPs, and the data are displayed as a heat map (Heatmap). Figure 5 shows that the DEPs screened by the standard 2.0-fold change threshold (with a fold change > 2.0 or <0.50, *p* < 0.05) can effectively separate the comparison groups, showing that the differentially expressed proteins screened can represent the difference between the two groups.

### 3.6. GO and KEGG Enrichment Analysis of the DEPs

A GO enrichment analysis was performed for both the up-regulated and down-regulated proteins in each comparison group to determine differences in the bioinformatic profiles of dairy cow placental proteins extracted using different proteases. The GO enrichment results showed differences in the biological information of placental proteins extracted using different proteases. The statistically significant (*p* < 0.05) network analysis performed using Cytoscape is shown in Figure 6. The DEPs identified using papain were enriched in collagen trimer, the extracellular region, the extracellular exosome, the chromaffin granule membrane, and protein heterodimerization (Figure 6A,B). The DEPs identified using trypsin were enriched in blood microparticles, membrane regions, a complex of collagen trimers, extracellular organelles, the extracellular matrix, extracellular matrix components, the cell surface, the viral nucleocapsid, the regulation of locomotion, the maintenance of location, syncytium formation, and anatomical structure formation involved in morphogenesis (Figure 6C,D). The DEPs identified using pepsin were enriched in the I band, immunological synapse, sarcomere, costamere, contractile fibers, structural molecule activity conferring elasticity, extracellular matrix binding, coagulation, the structural constituent of muscles, the NAD metabolic process, the structural constituent of the extracellular matrix, the collagen metabolic process, platelet activation, the regulation of plasma lipoprotein particle levels, and the cell death response in oxidative stress (Figure 6E,F).

To compare differences in the bioinformatic profiles of dairy cow placental proteins extracted using different proteases, KEGG enrichment analysis was performed for the up-regulated and down-regulated proteins of each comparison group. Protein digestion and absorption, glycolysis/gluconeogenesis, hypertrophic cardiomyopathy (HCM), dilated cardiomyopathy (DCM), focal adhesion, adherens junctions, tight junctions, leukocyte transendothelial migration, regulation of the actin cytoskeleton, the PI3K–Akt signaling pathway, focal adhesion, ECM–receptor interactions, regulation of the actin cytoskeleton, and amoebiasis were the common enriched pathways in each group (Figure 7). The unique enriched pathways among the DEPs produced by papain were the relaxin signaling pathway and the AGE–RAGE signaling pathway in diabetic complications (Figure 7A,B). The unique enriched pathways among the DEPs identified using pepsin were lysosomes, platelet activation, cardiac muscle contraction, bacterial invasion of epithelial cells, and small cell lung cancer (Figure 7C,D). The unique enriched pathways among the DEPs identified using trypsin were R and proline metabolism, olfactory transduction, proteasomes, protein processing in the endoplasmic reticulum, pyruvate metabolism, and arrhythmogenic right ventricular cardiomyopathy (ARVC; Figure 7E,F).

## 4. Discussion

Proteomics has been widely applied to study placenta-related diseases in humans, but few studies have investigated the placenta in dairy cows. Trypsin and pepsin are widely used in protein extraction and digestion in proteomics research [12,13,14,15], while papain is mostly used in the extraction of natural biologically active peptides [16,17]. In this study, based on the optimal enzymatic hydrolysis conditions of trypsin, pepsin, and papain for dairy cow placenta [8], this study further compared the detection of proteins after hydrolyzing the dairy cow placenta using a label-free MS method, and performed a bioinformatics analysis.

The comparison of the overall digestion efficiency of the three proteases showed that trypsin was superior to pepsin and papain regarding the number of proteins identified in the cow placenta. We presume that trypsin’s superior efficiency is due to its improved protein solubility and proteolytic efficiency. Trypsin has the advantage of high specificity, cleaving exclusively at arginine and lysine residues [6]. The optimal pH for trypsin activity is close to a neutral pH [18]—similar to the pH of the amniotic fluid, which is close to 7.0. Therefore, we assumed that placental proteins are soluble in a neutral environment, which may be the primary reason why trypsin had a better efficiency than pepsin and papain. Jiao et al. identified 788 proteins in the placenta of healthy pregnant mice using label-free proteomics and trypsin digestion [19]. Wawrzykowski et al. identified 886 proteins in cow placenta using two-dimensional separation and trypsin digestion [20]. Compared with the above-mentioned study, the amount of protein extracted using trypsin in the present study was relatively small; this may be due to the direct enzymatic hydrolysis of the placenta homogenate using trypsin, which lacked detergents and denaturants such as RapiGest. The protein digestion in the present study could be improved. However, the biological repetition rate of trypsin was >70%, the molecular weight and sequence coverage were relatively wide, the peptide lengths conformed to the characteristics of trypsin digestion, and the unique peptide distribution curve increased gradually. These findings indicate the reliability of the proteome data.

Pepsin also has strong specificity, cleaving peptides at aromatic and hydrophobic amino acids. Boukil et al. identified 110 proteins via the pepsin hydrolysis of mealworm meal [15]. This finding was similar to the present study findings, and the quality control data for the pepsin were better than those for trypsin and papain, indicating that pepsin can be reliably used in proteomics. However, the number of proteins identified using pepsin was significantly less than that identified when using trypsin, which may be due to the pH. The optimal pH range for pepsin activity is 1.5 to 2.0 [21]. However, some placental proteins have poor solubility in this acidic environment, which may lead to the low enzymatic hydrolysis efficiency of pepsin.

Papain is a protease that can release relatively larger quantities of bioactive peptides [22]. Therefore, it is mainly used for the extraction and digestion of natural active peptides. Oat proteins released using papain have a high ability to quench ABTS% + radicals and chelate ferrous ions while also displaying the second strongest activity for ROO% radicals [23]. The antioxidant activity of porcine liver hydrolysates using papain is relatively high [24]. Papain can release a large number of potential angiotensin converting enzyme (ACE)-inhibitory peptides from tilapia (*Oreochromis* spp.)-processing co-products, frame, and skin [16]. In the above-mentioned study, proteomic techniques were used to analyze the molecular characteristics of proteins. However, peptides with <5 amino acids (owing to their high Mws and electrical charge) are not effectively detected using MS/MS spectra [25]. Thus, we speculate that most of the peptides generated by papain hydrolysis have <5 amino acids, which leads to a high number of potential bioactive peptides and a small number of proteins that can be identified. The quality control data confirm this speculation, with a coefficient of variation of more than 15%, a low protein repetition rate and molecular weight, narrow sequence coverage and peptide length distribution, and few unique peptides.

Our findings indicate that the quantifiable proteins of cow placenta extracted using trypsin, pepsin, and papain were associated with almost the same biological processes, molecular function, and cellular component. The biological processes mainly included biological regulation and metabolic processes, similar to the protein patterns in the bovine placenta during early-mid pregnancy [26]. The cellular component included protein-containing complexes and the membrane, cytosol, nucleus, and extracellular space. The main cellular components for the identified proteins in retained and released bovine placenta are the cytoplasm, nucleus, and membrane [20]. This finding shows that the proteins that cause placental retention may be mainly distributed in the cytoplasm. Classification of the quantifiable proteins by molecular function revealed that the majority of proteins showed binding and catalytic activities, which is similar to the findings of Ner-Kluza et al. [26]. In conclusion, the GO annotation in the present study was quite similar to that in the literature. Furthermore, the KEGG analysis indicated that focal adhesion, the PI3K–Akt signaling pathway, human papillomavirus infection, and ECM–receptor interactions were the common pathways in cow placenta hydrolyzed using the three proteases. Proteins important for adhesive processes have been detected in the retained bovine placenta and disturbances in the metabolism of extracellular matrix proteins are speculated to lead to improper placental detachment. Our study confirmed that focal adhesion and ECM–receptor interactions were important for normally released placenta. 

Different enzyme digestion sites led to the identification of different proteins and consequently resulted in biological information differences. To explore these differences, the up-regulated and down-regulated DEPs were analyzed for each comparison group. Predictively, the results were completely different. Previous studies have shown that trypsin can be used in proteomic analyses of various samples, such as pork, beef, chicken, fish, milk, and shrimp [18,20,26,27,28]. Trypsin enzymatically decomposes mainly the extracellular matrix, blood particles, and cell-surface proteins. These proteins mainly perform positioning functions and participate in biological processes such as syncytium formation. Trypsin can be used for the study of amino acid metabolism, the proteasome and endoplasmic reticulum, dysosmia, and arrhythmogenic right ventricular cardiomyopathy. Pepsin enzymatically degrades mainly myofibrils, muscle fiber I-bands, actin ribs, and muscle contraction fibers; it has the function of activating elastic fibers and linking the extracellular matrix, and is involved in blood coagulation, muscle structure composition, NAD metabolism, and extracellular matrix composition. Oxidative stress leads to biological processes such as collagen metabolism and cell apoptosis. The above-mentioned findings indicate that pepsin has a significant hydrolysis effect on the muscle tissue in the placenta. Thus, more biological information related to the muscle tissue can be obtained using pepsin. Pepsin can be used to study diseases, such as those related to lysosomes, platelet aggregation, and myocardial contraction; bacterial invasiveness; and small cell lung cancer. Papain has a strong ability to decompose the extracellular matrix and collagen, and can be used to study the relaxin signaling pathway and the AGE–RAGE signaling pathway in diabetic complications, according to the GO and KEGG findings.

## 5. Conclusions

This study identified the differential proteins produced by the enzymatic hydrolysis of cow placenta using three different proteases. The production of these differential proteins is related to the characteristics of protease cleavage. In particular, proteins related to protein binding, ion binding protein-containing complexes, membrane metabolic processes, and biological regulation were detected in the enzymatic hydrolysis products. All in all, these findings highlight the basic biological information of the healthy dairy cow placenta and can be helpful for the screening of specifically expressed proteins and biomarkers in dairy cow pregnancy diseases. In addition, they can guide the selection of proteases for specific tissues and specific directions for proteomics.

## Figures and Tables

**Figure 1 animals-13-03395-f001:**
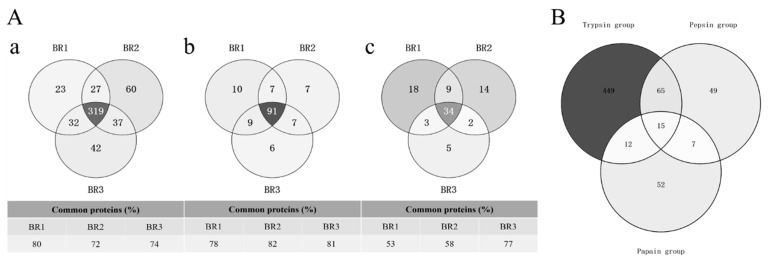
(**A**) Protein overlaps between replicates from cow placenta extracted by different proteases. Proteins obtained from cow placenta with trypsin (**a**), pepsin (**b**), and papain (**c**) were run in triplicate biological replicates. Data were processed using MaxQuant (version 1.5.3.17) with a protein FDR of 1%. (**B**) Identified common and unique quantifiable proteins between TRY, PEP and PAP. For each comparison, the total number of the quantified proteins per protease as well as common and unique proteins have been presented as number and percentage.

**Figure 2 animals-13-03395-f002:**
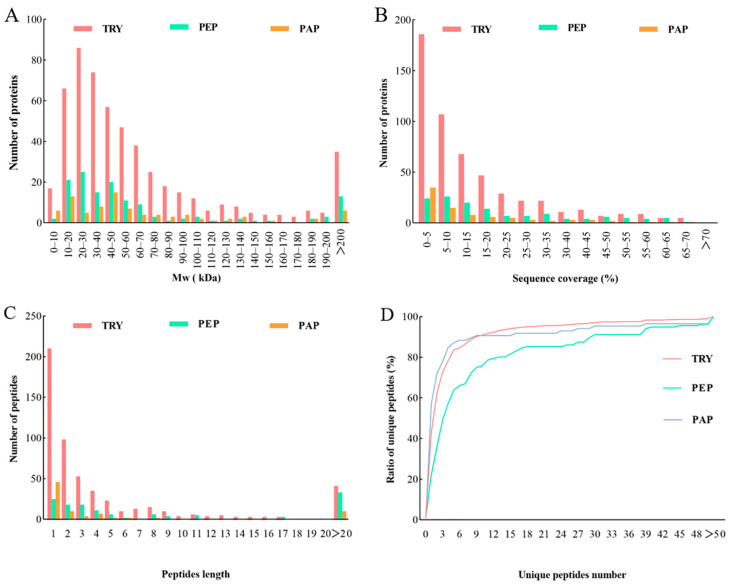
Qualitative and quantitative analysis of quantifiable proteins in cow placenta extracted with trypsin, pepsin, and papain. Analysis was performed based on the distribution of proteins according to: (**A**) molecular weight, (**B**) Sequence coverage, (**C**) Peptide length, (**D**) Unique peptides number.

**Figure 3 animals-13-03395-f003:**
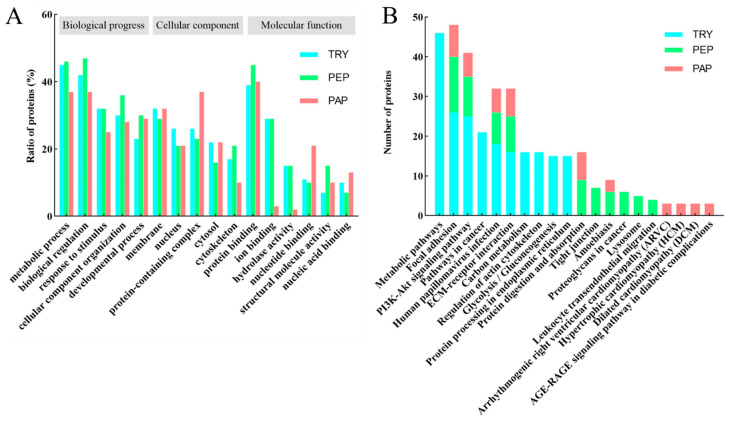
(**A**) Gene ontology (GO) analysis of quantifiable proteins. Functional assignments of proteins’ corresponding associated biological processes, molecular function, and cellular components are shown. The number of DEPs that could be assigned to the different categories is indicated. Blue column indicates TRY; green column indicates PEP; red column indicates PAP. (**B**) KEGG analysis of quantifiable proteins (Top 10). Blue column indicates TRY; green column indicates PEP; red column indicates PAP. Under the *x*-axis are KEGG terms; above each term is the number of quantifiable proteins.

**Figure 4 animals-13-03395-f004:**
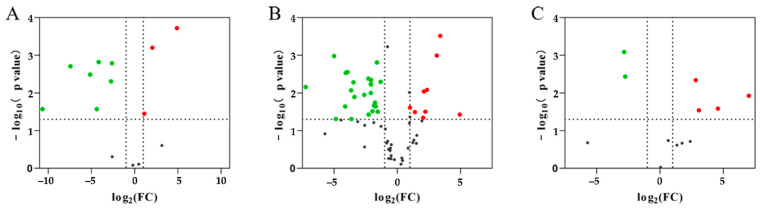
Volcano plot showing differentially abundant proteins between each group. (**A**) papain group vs. trypsin group; (**B**) pepsin group vs. trypsin group; (**C**) pepsin group vs. papain group. The *X*-axis is the negative log of the Fold-Change with 2 as the base, and the *Y*-axis is the negative log of the *p*-value with 10 as the base. Gray dots show no significant difference in a protein; red dots show significantly up-regulated proteins, and green dots show significantly down-regulated proteins.

**Figure 5 animals-13-03395-f005:**
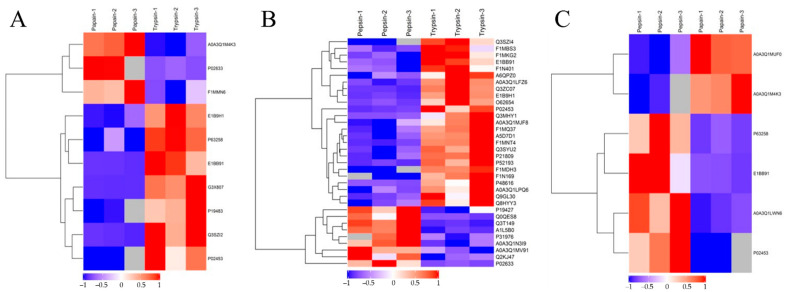
Cluster analysis of DEPs. Trypsin group vs. papain group (**A**); trypsin group vs. pepsin group (**B**); papain group vs. pepsin group (**C**). The hierarchical clustering results are represented by tree heat maps, in which each row represents a protein (that is, the ordinate is the protein with a significant difference), and each column represents a group of samples (Abscissa is the sample information). The logarithmic values (Log2Expression) of proteins with significant differences in expression in different samples are shown in the heat map in different colors, in which red represents significantly up-regulated proteins, blue represents significant down-regulated proteins, and gray represents the absence of protein quantitative information.

**Figure 6 animals-13-03395-f006:**
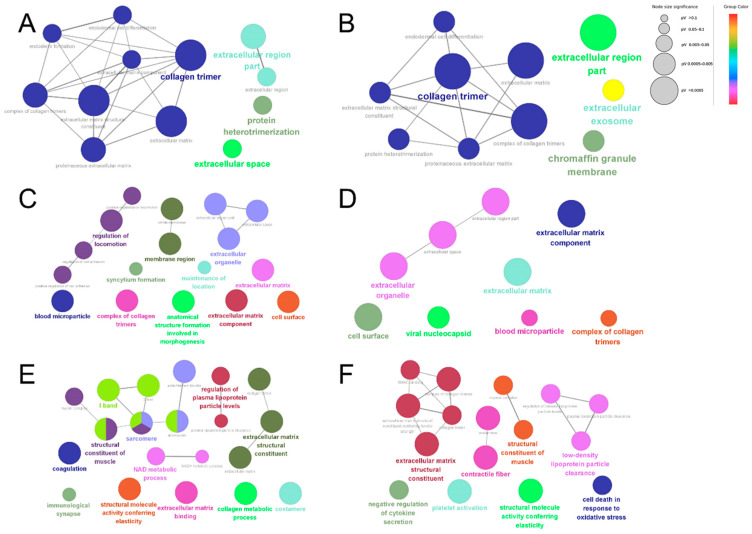
GO enrichment analysis of DEPs. Papain group vs. trypsin group up-regulated proteins (**A**); papain group vs. pepsin group down-regulated proteins (**B**); papain group vs. trypsin group down-regulated proteins (**C**); pepsin group vs. trypsin group down-regulated proteins (**D**); pepsin group vs. trypsin group up-regulated proteins (**E**); pepsin group vs. papain group up-regulated proteins (**F**). The node color indicates biologically similar reactions, and the size reflects the number of genes contributing to the pathway. If the reaction pathway shares 50% or more of the contributing genes, then they are connected by an edge. The representative nodes (based on FDR) are indicated by the colored texts (same below).

**Figure 7 animals-13-03395-f007:**
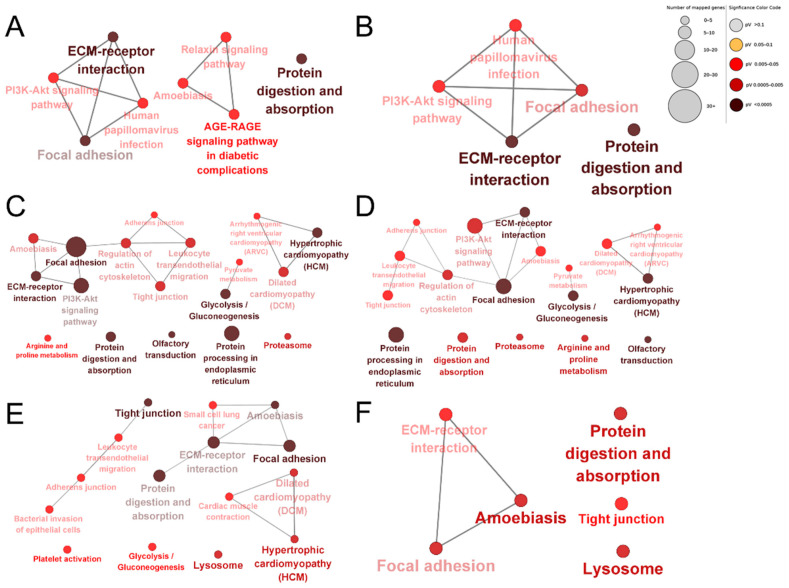
KEGG enrichment analysis of DEPs. Papain group vs. trypsin group up-regulated proteins (**A**); papain group vs. pepsin group down-regulated proteins (**B**); papain group vs. trypsin group down-regulated proteins (**C**); pepsin group vs. trypsin group down-regulated proteins (**D**); pepsin group vs. trypsin group up-regulated proteins (**E**); pepsin group vs. papain group up-regulated proteins (**F**). The shade of colors indicates the size of the *p*-value.

**Table 1 animals-13-03395-t001:** Protein and peptide identifications from cow placenta.

Sample Name	Proteins						Peptides					
	BR 1	BR 2	BR 3	Mean	SD	CV (%)	BR 1	BR 2	BR 3	Mean	SD	CV (%)
TRY	401	443	430	425	18	4	2022	2489	2002	2171	225	10
PEP	117	111	113	114	2	2	993	939	816	916	74	8
PAP	64	59	44	56	8	15	99	98	62	86	17	20

The table lists numbers of identified peptides and proteins in each biological replicate, together with the average number of identification (Mean ± SD) and coefficient of variation (CV%). There are three groups of biological replicates (BR), namely BR1, BR2, and BR3.

**Table 2 animals-13-03395-t002:** Specially expressed proteins in each comparison group.

Comparisons	Consistent Presence/Absence Expression Profile	
	Presence	Absence
Papain group vs. Trypsin group	31	391
Trypsin group vs. Pepsin group	47	340
Pepsin group vs. Papain group	96	34

“Presence” refers to proteins with consistent presence in the first group and absence in the other group; " Absence " refers to proteins with consistent presence in the second group and absence in the other group.

## Data Availability

Mean values of all data generated or analyzed during this study, mass spectrometry parameters, and detailed identification for proteins are included in this published article and its additional information files. Individual data are available from the corresponding author on reasonable request.

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
