# Peer review of "Analysis and Comparison of Proteomics of Placental Proteins from Cows Using Different Proteases"

_animals, 2023, doi:10.3390/ani13213395_

Round 1

Reviewer 1 Report

Comments and Suggestions for Authors

The present manuscript is a report on the comparative efficiencies of separation of proteins in cow placentas by three proteases: trypsin, pepsin, and papain. The study is completed thoroughly and leads the authors to conclude that trypsin is the best-suited protease for digesting cow placenta.  The results are suitable for other researchers to utilize. However, the manuscript needs to be edited by an English-speaking person or translation service to improve the flow of the text and correct use of English grammar. 

Comments on the Quality of English Language

The manuscript needs to be edited by an English-speaking person or translation service to improve the flow of the text and correct use of English grammar. This is important for complete sentences throughout that will increase the readability of the manuscript.

Reviewer 2 Report

Comments and Suggestions for Authors

In this manuscript, Shen and colleagues compare the proteome of cow placental generated by three proteases and obtain some interesting results. However, there are several concerns that should be addressed.

#Why enzyme reaction time is different for the three proteases? 

#What is the peptide yield by each protease? How much peptide was used for LC-MS injection?

# In MaxQuant searching, carbamidomethyl was used as fix modification. Did the authors perform 2-Iodoacetamide blocking? If yes, the information should be provided in Methods part.

# The authors call it “protein extraction” (line 138). Is it a proper description? The product after protease digestion is peptide, not protein. 

# In Discussion, authors describe the previous application of trypsin and pepsin in proteomics analysis of placenta. Since they had been already published, why the authors design this experiment? How the authors will use their new information for future study of placenta?

Reviewer 3 Report

Comments and Suggestions for Authors

The work of Shen et. al. with the title "Analysis and Comparison of Proteomics of Cow Placental Proteins Using Different Proteases".  Brings a methodological paper, with a great interest for the general animal science community. Basically what the author proposes is to evaluate the effect on the detection of protein by MS using different proteases. The tissue of choice was the placenta, which is a very interesting tissue. Since the paper is based on method's effect on results, one should expect a highly standardized and well-described paper. Surprisingly, the Material and methods are not reaching publishing standards. 
Protease details of the brand and catalog number should be given.
The authors analyzed 9 placentas, obtained after delivery, after how much time from the delivery to placentome collection?
On table 1 there is BR1 to BR3, were 3 or 9 samples? This makes the reader confused, not all placentomes were used. The samples from each treatment were different? Different animals, placentas, for each treatment, or all the same placentomes were used in all groups? This issue must be 
The other MM section must be improved and well described in detail. 
Results:
Table 1 and Figure 1, Is confusing, the MM says 9 animals, this results has one 3 animals, why?
I suggest to combine Figure 1 and Figure2  to make one panel/Figure. Same thing for Figure 4 and Figure 5.
Figure 7 is too small. 
 Results and Discussion is ok, but should verified for minor grammatical and spelling errors.
Conclusion end with "At the same time, it can provide guidance on the selection of protease for specific tissues and specific directions for proteomics". But it is not stated when one should use one of the proteases , conclusion should be more bold and state when one should be used instead of the other.

It is an interesting result, that can be useful as a guide for several experiments once all the issues mentioned are corrected. I can not recommend this work for publishing in its present form, but I am keen to read this work again after revision.

Round 2

Reviewer 2 Report

Comments and Suggestions for Authors

I have no concerns about the paper.